# Bipolar dispersal of red-snow algae

Takahiro Segawa [1,2], Ryo Matsuzaki[3], Nozomu Takeuchi [4], Ayumi Akiyoshi[2], Francisco Navarro [5], Shin Sugiyama[6], Takahiro Yonezawa[7,8] & Hiroshi Mori [9]

Red-snow algae are red-pigmented unicellular algae that appear seasonally on the surface of thawing snow worldwide. Here, we analyse the distribution patterns of snow algae sampled from glaciers and snow patches in the Arctic and Antarctica based on nuclear ITS2 sequences, which evolve rapidly. The number of phylotypes is limited in both polar regions, and most are specific to either the Arctic or Antarctica. However, the bipolar phylotypes account for the largest share (37.3%) of all sequences, suggesting that red-algal blooms in polar regions may comprise mainly cosmopolitan phylotypes but also include endemic organisms, which are distributed either in the Arctic or Antarctica.

[1] Center for Life Science Research, University of Yamanashi, 409-3898 1000 Shimokato, Chuo, Yamanashi, Japan. [2] National Institute of Polar Research, 10-3 Midori-cho, Tachikawa, Tokyo, Japan. [3] Center for Environmental Biology and Ecosystem Studies, National Institute for Environmental Studies, 16-2 Onogawa, Tsukuba, Ibaraki, Japan. [4] Department of Earth Sciences, Graduate School of Science, Chiba University, 1-33 Yayoi-cho, Inage-ku, Chiba, Japan. [5] Departamento de Matemática Aplicada a las Tecnologías de la Información y las Comunicaciones, ETSI de Telecomunicación, Universidad Politécnica de Madrid, Av. Complutense, 30, 28040 Madrid, Spain. [6] Institute of Low Temperature Science, Hokkaido University, Nishi8, Kita19, Sapporo, Japan. [7] Department of Animal Science, Faculty of Agriculture, Tokyo University of Agriculture, 1737 Funako, Atsugi, Kanagawa, Japan. [8] School of Life Sciences, Fudan University, SongHu Rd. 2005, Shanghai 200438, China. [9] Center for Information Biology, National Institute of Genetics, 1111 Yata, Mishima, Shizuoka, Japan. Correspondence and requests for materials should be addressed to T.S. (email: tsegawa@yamanashi.ac.jp)

Red snows are a worldwide phenomenon during the melt season and are caused by blooms of red-pigmented green algae (Chlorophyceae) in thawing snow. Historically, they have been recorded in the daily logs of polar and alpine explorers such as Captain John Ross and Charles Darwin[1,2] (Fig. 1). The red pigments are carotenoids that serve as antioxidants, as an energy sink, and as a light shield for algal cells exposed to the intense radiation, particularly to the photosynthetically active radiation (PAR), on the snow surface[3–5]. The algae photosynthetically produce organic matter, which can reduce the snow-surface albedo and accelerate the melt rate, and thus algae have an impact on cryospheric environments[6,7].

Several taxa of red-snow algae have been recognized in snow fields worldwide, and most have been identified based on microscopic features of the cells. Spherical red-snow cells have often been identified as *Chlamydomonas* cf. *nivalis*[8] and can be regarded as a cosmopolitan cryophilic species. On the other hand, several studies have used next-generation sequencing technology to elucidate the geographic distribution pattern of red-snow algae based on molecular data[9,10]. Recently, red-snow algae collected from different regions of the Arctic were reported to be cosmopolitan based on 18S rRNA gene analysis[11]. This gene has frequently been used for establishing algal taxonomy at the species level[4,12]. However, the 18S rRNA gene resides within a relatively

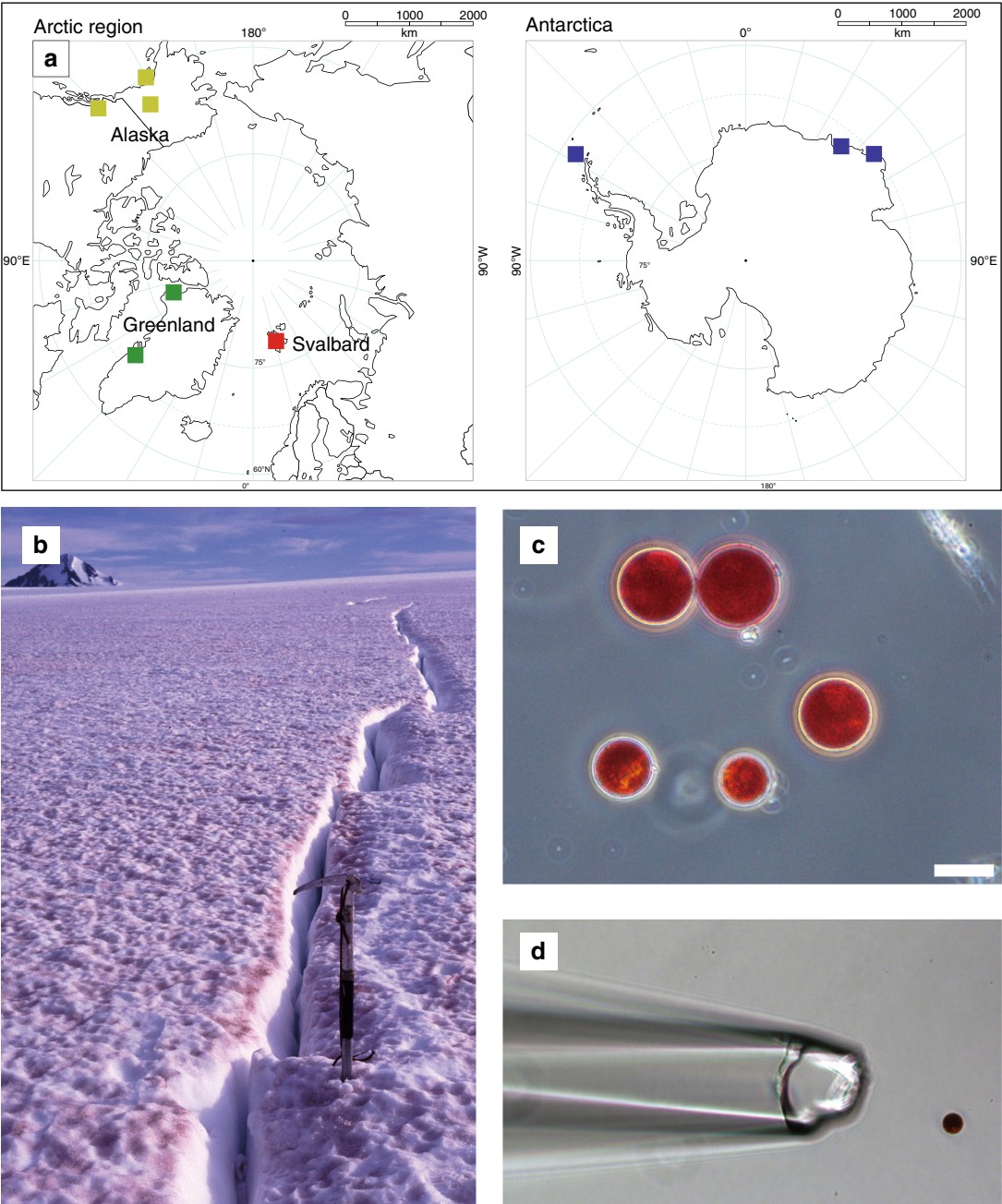

**Fig. 1** Sampling sites and red-pigmented green algae causing red snow phenomena. **a** Map showing the locations of red-snow sampling sites. **b** Representative photograph of red snow on Harding Icefield in Alaska. **c** Microscopic view of the dominant snow algae contained in the red-snow samples. Scale bar, 20 μm. **d** An isolated cell of a red-snow alga used for single-cell PCR. To confirm that the sequences were obtained from red-snow algae, single cells from the red-snow samples were manipulated by micromanipulator system. Maps in panel A sourced from MAPIO WORLD, DesignEXchange Co. Ltd (Copyright 2005)

slow-evolving region, and therefore we do not feel it is very suitable for elucidating the geographic distribution of algae—especially the interpretation of global distribution—because the resolution of any phylogeographical analysis depends on evolutionary rates of focusing genes.

Here, we analyse the geographical distribution of snow algae on red snows collected from the Arctic and Antarctica, using the sequences of the nuclear rDNA internal transcribed spacer 2 (ITS2) region. ITS2 has a high evolutionary rate and is thus suitable for revealing fine-scale genetic structures. We show that some red-snow algal phylotypes have a bipolar distribution, whereas others are present in only limited areas.

## Results

**Classification of red-snow algae by 18S rRNA gene and ITS2.** We used high-throughput sequencing and single-cell PCR to analyse the geographical distribution of snow algae that cause red snow at polar glacier surfaces and seasonal snow fields on non-glaciered area (omitting mid-latitude mountainous regions). Our analysis focused on unique sequences (phylotypes) of red-snow algae at 24 red-snow sites in the Arctic and Antarctica (Fig. 1 and Supplementary Table 1). In total, 64,047 unique sequences for algal ITS2 were found in all the red-snow samples, and 348 operational taxonomic units (OTUs) were defined with ≥98% nucleotide sequence identity. In the ITS2 sequence, only 38 of 348 OTUs (accounting for 1.2% of the total sequencing reads) were >95% similar to the culturable strains in the NCBI-nt database, indicating that the majority of red-snow algae strains for which an OTU was identified have yet to be successfully cultured. Because information on algal ITS2 sequences is currently limited, it is difficult to conduct taxonomic classification of sequences based on general homology-based search approaches such as the BLAST algorithm using only a public DNA sequence database. Therefore, we established an in-house red-snow ITS2 sequence database (http://redsnow18.paleogenome.jp). For this purpose, the 18S-ITS sequences were obtained from red-snow samples via Sanger sequencing; additionally, some sequences were obtained directly from cells that had been morphologically identified as *C. nivalis* from the red-snow samples via single-cell PCR.

We classified the ITS2 sequences at the species level according to the generic species concept based on secondary structural differences in the ITS2 region, which correlate with the delimitation of biological species[13]. We confirmed that the predicted ITS2 secondary structures of OTUs have four helices, a U–U mismatch in helix II, and a YGGY motif on the 5′ side near the apex of helix III, all of which are common structural hallmarks of ITS2 in eukaryotes[14,15]. The ITS2 sequences were classified into 22 subgroups belonging to five chlorophycean and six trebouxiophycean groups based on their secondary structures and BLASTN results (Fig. 2a, Supplementary Figs. 1–21). Among those 22 subgroups, sequences belonged to the 'Chlamydomonas'-snow group A (6%), 'Chlamydomonas'-snow group B (71%), *Raphidonema* group (16%), and *Chloromonadinia*-snow group G (6%). The 'Chlamydomonas'-snow group included field-collected samples from Svalbard and the European Alps that had been assigned to *C. nivalis* (Supplementary Fig. 22). In addition, several studies have reported *Chloromonas* in the *Chloromonadinia*-snow group as identified from glacier snow samples[11,16]. The aforementioned four groups accounted for 99% of the total number of sequences analysed, and hence they can be considered the major groups present on red snow in the Arctic and Antarctica (Fig. 2b, Supplementary Tables 2–4).

**Endemic distribution of most red-snow algal phylotypes.** The composition of the snow algal communities was not homogenous across the sample sites, and the predominant groups of snow algae also differed between regions although the Shannon-Wiener diversity index did not differ significantly between regions ($P > 0.05$; Supplementary Figs. 23 and 24). The phylogenetic composition of the algae communities differed significantly among regions, and indeed the genetic structure differed between the Arctic and Antarctic samples (Fig. 2b, Supplementary Figs. 25–27, Supplementary Tables 5 and 6). In addition, the snow algal communities could be correlated with geographical distances between the Arctic and Antarctica. This correlation also suggested that the snow algal population could be distinguished based on geographic location (Supplementary Table 7). The genetic differentiation between the Arctic and Antarctica seemed to be related to the ability to migrate rather than the adaptability of their genotypes to various snow environments. These results suggest the existence of a geographic barrier for algae dispersal between the Arctic and Antarctic environments.

We found that most of the algal OTUs are endemic to a particular polar region. In our analysis of the 22 subgroups based on ITS2 secondary structural differences, 15 subgroups were found to be endemic, i.e., were detected in either the Arctic or Antarctica; these 15 subgroups were of low abundance, however, accounting for only 6.0% of the total sequencing reads. In addition, based on the analysis of the 64,047 ITS2 unique sequences, an average of 55.1% of the unique sequences were endemic to a particular region (Antarctica, 77.9%; Svalbard, 49.9%; Greenland, 21.7%; Alaska, 70.8%), accounting for 21.4% of the total sequencing reads (Fig. 3a, Supplementary Tables 8–10). This result suggests that the major unique sequences are endemic based on their distribution, although they apparently do not contribute in a major way to the total sequencing reads. A previous study based on an 18S rRNA gene analysis concluded that cryospheric algae are commonly found in different regions of the Arctic[11]. However, we found that only 1.9% of unique sequences were commonly distributed for the entire Arctic group (distributed across all Arctic regions but absent from Antarctica; Supplementary Table 11). Our results using ITS2 sequences revealed that approximately one-half of the unique sequences were regionally endemic, suggesting limitations to dispersal (Fig. 3a).

**A few bipolar phylotypes of snow algae predominate on red snows.** We carried out a detailed analysis of the bipolar distribution patterns of snow algae in the red-snow samples from the Arctic and Antarctica. We found that only limited algal unique sequences are distributed in both polar regions, but they account for a large proportion of the sequencing reads based on analyses of the ITS2 region and 18S rDNA (Fig. 3a, b, Supplementary Figs. 28–30). A geographic network analysis of the ITS2 region revealed that almost all unique sequences were distributed regionally, and only a few unique sequences were found among the cosmopolitan species or at multiple sites (Fig. 3c). We observed only 912 unique sequences for the cosmopolitan distribution classified as the 'Chlamydomonas'-snow group B and *Raphidonema* group. These 912 unique sequences were of low abundance (3–9%, depending on region), but they accounted for a large proportion of the sequencing reads (average 37.3%; Antarctica, 12.3%; Svalbard, 47.8%; Greenland, 68.4%; Alaska, 27.2%; Fig. 3b). A total of 62 and 850 cosmopolitan unique sequences were classified as 'Chlamydomonas'-snow group B and *Raphidonema* group, respectively, accounting for 29 and 8% of the total sequencing reads (Supplementary Table 12 and Supplementary Fig. 31). A limited number of snow algal unique sequences in these two groups was distributed in a bipolar manner, and the sequences were globally dispersed across both polar regions and

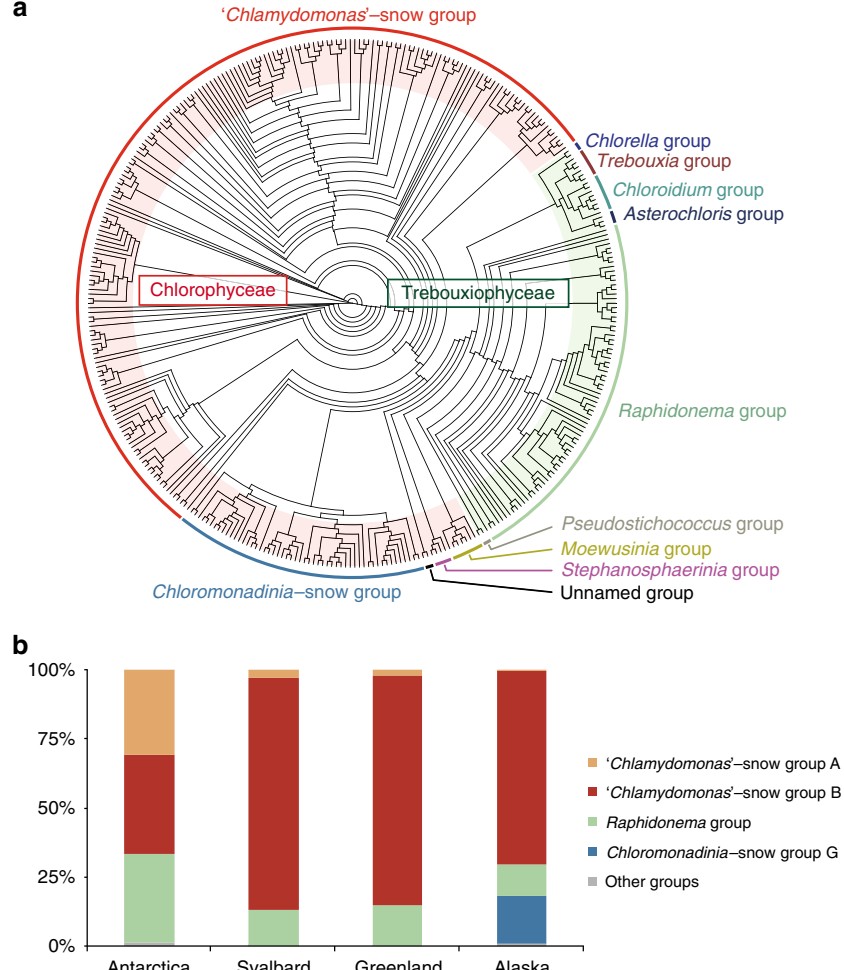

**Fig. 2** Classification of red-snow algae based on ITS2 sequence. **a** Phylogenetic relationship of 348 OTUs (98% OTU) based on ITS2 region sequences. A maximum likelihood tree was constructed with 1000 bootstrap replications using RAxML version 8.2.10 with the GTR+I+Γ model. In 98% OTUs, identical sequences were reduced to a single OTU. Taxonomic groups that were defined by the secondary structure of ITS2 are labelled and distinguished based on colour. For detailed information, see Supplementary Figure 1. **b** Algal taxonomic composition of the Illumina reads in red-snow samples based on ITS2 sequences. The bar chart presents data for the average community composition of each region based on the four major ITS2 groups and low-abundance groups (grouped as "Other groups")

predominated on red snows. For the *Raphidonema* group, certain species inhabit soil environments, and the secondary inhabitants of red snows may be a consequence of having been wind-blown from distant snow surfaces[17]. A future analysis of the communities in the environment adjoining red snows (e.g., soils) will be important for understanding the causes underlying the global dispersion of the two groups.

## Discussion

A current hot topic is whether the biogeographical distribution of microorganisms is global or local. The microbial cosmopolitan dispersion hypothesis of Baas Becking ("Everything is everywhere, but the environment selects")[18] is often invoked to explain the observed patterns of global algae distribution driven by the capacity for widespread dispersal. Our results suggest that a few cosmopolitan species of green algae dominate red-snow habitats, but indeed much endemism was detected. Our findings underscore the importance of understanding the ecology of snow algae as well as improving the population analyses and taxonomic classification methods that utilize environmental samples. Recent work based on globally sampled 16S and ITS sequence data has demonstrated the spatial distribution patterns of cryosphere

microorganisms as well as their regional differentiation and adaptability to environments using glacial cyanobacteria samples acquired from both polar and mid-latitude mountain regions[19,20]. To understand the mechanism by which snow algae form geographically specific population structures and how they migrate across the global cryosphere, it will be necessary to study samples from glaciers and snowpacks in mid-latitude mountain ranges such as those in Europe, Asia, and the Americas, where red-snow algae also commonly bloom.

## Methods

**Samples**. We used red-snow samples collected during the melt season from 24 sites on 13 glaciers and seasonal, non-glacier snow fields in the Arctic (including Greenland, Svalbard, and Alaska) and Antarctica (including Livingston Island, the Riiser-Larsen and Yukidori Valley; Supplementary Table 1). Red-snow samples were collected in sterile 50-mL plastic conical tubes or plastic bags (Whirl-pak, Nasco, USA). Samples were kept frozen during transport to the National Institute of Polar Research (Tokyo, Japan) and then stored at –80 °C before use.

**DNA extraction**. Red-snow samples were melted at 4 °C, and 5–10 mL of each sample was centrifuged at $5000 \times g$ for 10 min to obtain a pellet. The pellets from five replicate samples collected at each site were pooled and used for DNA extraction. Genomic DNA was extracted from each pellet using a FastDNA spin kit for soil (Qbiogene, USA) with a Yasui Kikai (Japan) Multi-beads shocker at 2500

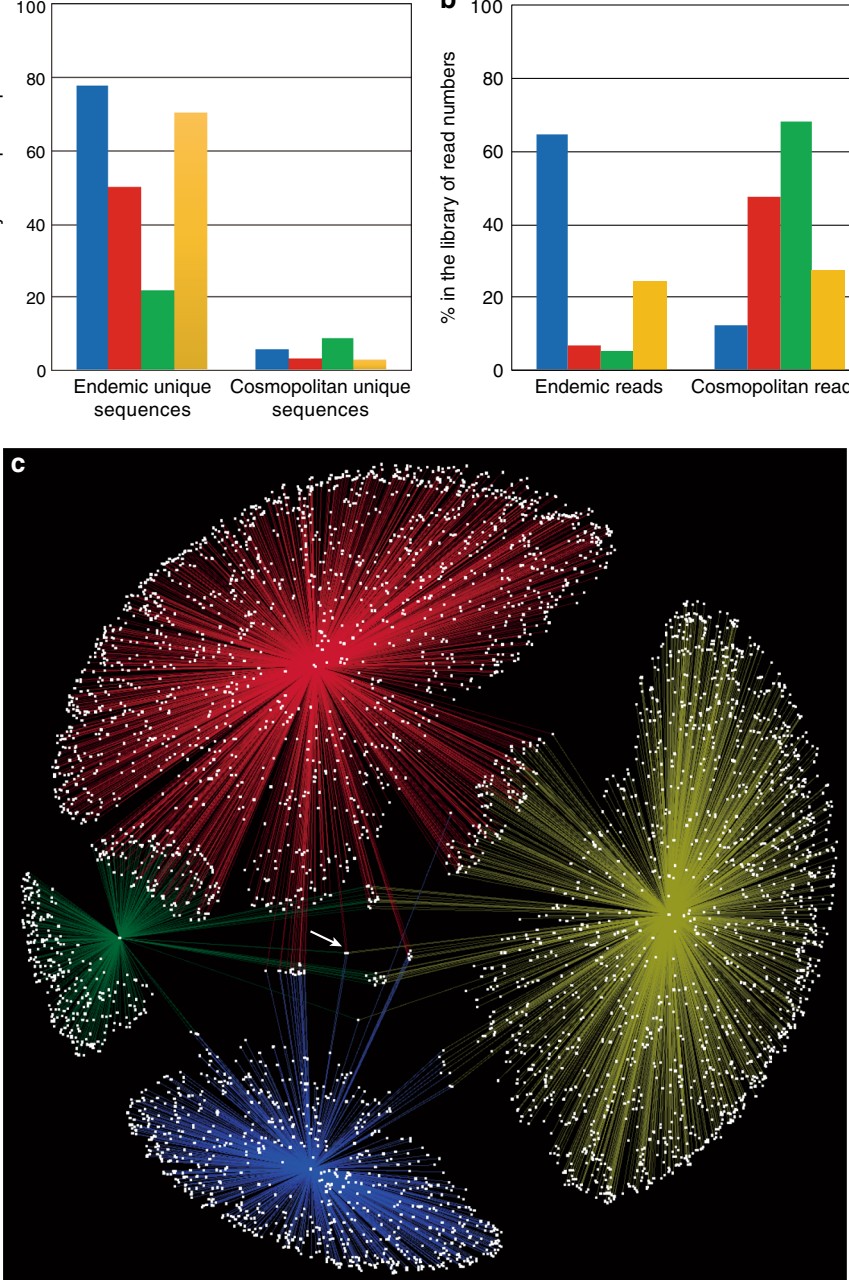

**Fig. 3** Distribution of the endemic and bipolar cosmopolitan snow algae obtained from each region based on ITS2 unique sequences from high-throughput sequencing. OTUs and sequencing read numbers are shown. **a** Unique sequences. **b** Sequencing read numbers of the unique sequences. Bars are coloured according to geographical region of the glacier: blue, Antarctica; red, Svalbard; green, Greenland; yellow, Alaska. **c** Co-occurrence networks for ITS2 unique sequences of snow algae from both polar regions. Unique sequences are coloured according to geographical region of the glacier, as noted above. The arrow points to the cosmopolitan-distributed phylotypes

rpm for 30 s. All DNA extractions were conducted on a class 100 clean bench (Sanyo, Japan), and subsequent procedures were carried out on another class 100 clean bench.

**18S rRNA–ITS2 long-read strategy by Sanger sequencing**. We constructed our in-house red-snow ITS2 sequence database in this study. Because information on algae ITS2 sequences is limited in current public DNA sequence databases, it is difficult to conduct taxonomic classification based on general homology-based search approaches. For constructing the in-house database, contiguous 18S–ITS2 sequences were obtained from red-snow samples by Sanger sequencing. The 18S rRNA–ITS2 region sequences were obtained from red-snow samples collected from three glaciers in Alaska (Harding Icefield, Gulkana glacier, Juneau ice field), one in Svalbard (Austre Brøggerbreen), and one in Antarctica

(Riiser-Larsen). PCR amplification of the eukaryotic 18S rRNA gene to the ITS2 region was performed using high-fidelity PrimeStar polymerase (Takara, Japan) and primers EUK F (AACCTGGTTGATCCTGCCAGT)[21] and ITS4 (TCCTCCGCTTATTGATATGC)[22]. To reduce PCR artifacts, the number of PCR cycles was kept to a minimum. The PCR conditions were as follows: initial denaturation at 94 °C for 3 min, then 12–20 cycles at 98 °C for 10 s, 57 °C for 15 s, and 72 °C for 4 min, with final extension at 72 °C for 7 min. To minimize PCR drift, three independent reactions were conducted. The pooled PCR products were purified using the MinElute PCR Purification kit (Qiagen, Germany) and cloned using the pCR4Blunt-TOPO with a Zero Blunt TOPO PCR Cloning kit for Sequencing (Life Technologies, USA). *Escherichia coli* HST08 Premium Competent Cells (Takara) were transformed with the cloning vector. We sequenced approximately 1100 clones in total using Big Dye Terminator 3.1 and an ABI 3130xl automatic sequencer.

**Single-cell PCR and Sanger sequencing analysis**. To obtain the sequences from algal cells morphologically identified as *C. nivalis* from the red-snow samples, we carried out a single-cell PCR approach. Single-cell PCR was used to directly obtain 18S rRNA–ITS2 region sequences from snow algal cells contained in the red-snow samples, which were collected from three glaciers in Alaska (Harding Icefield, Gulkana glacier, Juneau ice field) and one in Svalbard (Austre Brøggerbreen). Single cells of snow algae that carried red pigments were isolated using a CKX41 microscope (Olympus, Japan) and a micromanipulator system (Eppendorf, Germany) on a class 100 clean bench. Single cells were transferred into 0.2-mL PCR tubes (Eppendorf) and washed twice with distilled water that had been sterilized by UV irradiation, and then each algal cell was transferred into a PCR tube containing a sterilized lysis buffer (10 mM Tris-HCl pH 8.0, 0.1 mM EDTA, 0.1% Tween 20 in MilliQ water). Tubes containing algal cells were subjected to three consecutive freeze-thaw cycles to lyse the cell wall and then incubated at 55 °C for 2 h. Each lysate was subjected to PCR amplification of the 18S rRNA–ITS2 region using Ex Taq Hot Start Version (Takara) with the primers P1 (ATCTGGTT GATCCTGCCAGT)[23] and ITS4[22] under the following conditions: initial denaturation at 94 °C for 3 min, followed by 45 cycles of 94 °C for 45 s, 57 °C for 45 s, and 72 °C for 4 min, with final extension at 72 °C for 7 min. Sequencing procedures were the same as described in above. We used more than 20 cells in each sample for single-cell PCR, and we successfully determined the sequences from 10 cells in total.

**18S rRNA gene and ITS2 region analysis by Illumina sequencing**. Partial 18S rRNA gene sequences and ITS2 sequences were amplified using primers TAR euk454FWD1 (CCAGCASCYGCGGTAATTCC) / TAReukREV3 (ACTTTCGTTCTTGATYRA)[24] and Coleman c (GCATCGATGAAGAACGC AGC) / b (GGGGATCCATATGCTTAAGTTCAGCGGGT)[25], with Illumina overhang adaptor sequences attached to their 5′ ends, respectively. The overhang adapter sequences TCGTCGGCAGCGTCAGATGTGTATAAGAGACAG and GTCTCGTGGGCTCGGAGATGTGTATAAGAGACAG were added to the 5′ end of the forward and reverse primers, respectively. Each reaction (25 µL) contained 1 × KAPA HiFi HS ReadyMix (Kapa Biosystems, USA), 0.2 µM of each primer, and 2 µL of template DNA. PCR was performed under the following cycling conditions: initial annealing at 95 °C for 3 min, followed by 15–20 cycles at 95 °C for 30 s, 50 °C for 30 s, and 72 °C for 60 s, and a final extension at 72 °C for 5 min. The amplicons with Illumina overhang adapter sequences were generated in triplicate, and each amplicon was pooled before index PCR. The PCR products were labelled with two sample-specific indices that each contained a sample-unique index[26] and Illumina adapter sequences at their 5′ end (Nextera XT index kit v2, Illumina). Each PCR mixture (10 µL) contained 1 × KAPA HiFi HS ReadyMix (Kapa Biosystems), 2 µL each of forward and reverse primers, and 1 µL of the recovered PCR products. PCR was performed under the following cycling conditions: 95 °C for 3 min, followed by 8 cycles of 95 °C for 30 s, 55 °C for 30 s, and 72 °C for 60 s, with final extension at 72 °C for 5 min. After agarose gel electrophoresis, PCR products were excised from the gel and purified using a NucleoSpin Gel and PCR clean-up kit (Macherey Nagel, Germany). Tagged amplicons were pooled for high-throughput sequencing with the Illumina MiSeq platform in 300-bp paired-end sequencing reactions with v3 reagent kit (Illumina). To avoid index switching with MiSeq[26], we carried out two separate MiSeq runs for the Arctic and Antarctic samples.

**Sequence quality filtering**. We merged the forward and reverse MiSeq reads for the paired-end libraries of ITS2 and the 18S rRNA gene using USEARCH (version 7.0.1090)[27] with the fastq_truncqual parameter. Only paired reads carrying the exact index combinations were assigned to the sample's reads and used for subsequent analyses. We then discarded the reads that (i) contained ambiguous nucleotides, (ii) contained <50 nt, and (iii) were mapped to the PhiX genome sequence by a Bowtie 2 (version 2.2.4) search with default parameters[28]. The forward primer, reverse primer, and adapter sequences were removed by a TagCleaner (version 0.12) search allowing up to four mismatches[29]. The unique ITS2 and 18S rRNA gene sequence clusters, i.e., the non-redundant sequence clusters, were constructed by clustering high-quality reads using UCLUST (version 7.0.1090)[27]. Chimeric clusters were detected and removed if the unique ITS2 or 18S rRNA gene sequence clusters were assigned to the chimera in UCHIME in de novo mode (version 7.0.1090)[30]. We also discarded unique singleton sequence clusters for which the unique clusters consisted of only one read. Then, 98% sequence clustering of unique ITS2 sequence was conducted with UCLUST (version 7.0.1090) with identity >98% and query and reference coverage >80%.

**Taxonomic assignments of ITS2 and 18S rRNA gene sequences**. Before conducting a taxonomic assignment of the ITS2 sequences, we constructed an in-house red-snow ITS2 sequence database using 18S–ITS2 sequences obtained from Sanger sequencing of the PCR products of the clone library and single cells. We manually extracted the 18S rRNA gene region from the 18S–ITS2 sequences. To obtain sequences similar to our 18S rRNA gene sequences in the National Center for Biotechnology Information (NCBI) database (https://www.ncbi.nlm.nih.gov/), we initially performed a BLASTN search[31]. Our Sanger sequencing based

18S rRNA gene sequences and public 18S rRNA gene sequences were aligned using the MAFFT[32]. Maximum likelihood trees were reconstructed using RAxML v.8.2.10[33] with the GTR + Γ models. Using the 18S rRNA gene phylogenetic tree information, we picked the 18S–ITS2 sequences that could belong to the 'Chlamydomonas'-snow group and Chloromonadinia-snow group. We extracted the ITS2 region from the 18S–ITS2 sequences belonging to these two groups. We then conducted a sequence clustering with a 99% nucleotide identity level based on the ITS2 sequences in each sample, using the furthest neighbour algorithm in Mothur 1.38.1[34]. The representative sequences of the ITS2 sequence clusters from these two groups were named as "in-house red-snow ITS2 sequence database" (http://redsnow18.paleogenome.jp).

Taxonomic assignments of unique ITS2 sequences were conducted by a BLASTN search with a top-hit E-value <1e$^{-8}$, identity >90%, and alignment length >200 bp against (i) the UNITE fungal ITS2 sequence database[35], (ii) Viridiplantae ITS2 sequences obtained from NCBI, and (iii) an in-house red-snow ITS2 sequence database. Taxonomic assignments of unique 18S rRNA gene sequences were conducted by a BLASTN search with a top hit E-value <1e$^{-8}$, identity >90%, and alignment length >150 bp against the SILVA SSU Ref NR99 sequence database (version 123;[36] extended with additional 4 sequences of cryophilic algae, Chloromonas nivalis (AF514409), Chloromonas tughillensis (AB906348), Raphidonema nivale (AF448477), and Raphidonema sempervirens (AF514410)).

**Groups based on the secondary structure of the ITS2**. For estimating the diversity of microalgae in the red-snow samples, we classified the ITS2 sequences with the MiSeq analysis at the species level according to the generic species concept based on structural differences in ITS2[13]. Hereafter, we define these unique sequences as "phylotypes" and 98% sequence clusters from unique ITS2 sequence as "OTUs". The 5.8S-28S rRNA interaction region within the sequence of each OTU (Supplementary Figs. 3–21) was annotated using the web interface for Hidden Markov Models–based annotation[37] at the ITS2 database[38]. Secondary structures for ITS2 were predicted using Centroidfold[39] and RNAfold WebServer[40] and were manually refined. We confirmed that the ITS2 secondary structures of the OTUs examined in this study contained four helices, a U–U mismatch in helix II, and a YGGY motif on the 5′ side near the apex of helix III, which are common structural hallmarks of those of eukaryotes[14,15]. The ITS2 sequences of the OTUs were then compared with published sequences using BLASTN in the NCBI database. Based on the BLASTN results, the OTUs were classified into five chlorophycean and six trebouxiophycean groups (Supplementary Fig. 2). Within each group, species boundaries among the OTUs were estimated based on the compensatory base change near the apex of helix III encompassing the YGGY motif in the ITS2 secondary structure (the most conserved region of the ITS2 secondary structure of eukaryotes[41]; the compensatory base change correlates with the separation of biological species[13].

**Statistical analyses of algae ITS2 sequences**. The unique sequences and OTUs of the algal ITS2 sequence clusters were identified from the taxonomic assignment results, and the remaining unique sequences and OTUs of ITS2 that were not assigned to algae were discarded. The hierarchical clustering of samples, as by calculating the Bray-Curtis dissimilarity index of unique and OTU compositions, was conducted with R (version 3.2.4) and a VEGAN library. Pairwise differences among the four regions (Antarctica, Svalbard, Alaska, and Greenland) of unique and OTUs of ITS2 sequence cluster compositions were statistically analysed with PERMANOVA. Shannon-Wiener indices for snow-algae communities in the various geographic regions were calculated using R software. Correlations between the regional physical distance between sample sites and the Bray-Curtis dissimilarity index of unique or OTUs of ITS2 sequence cluster compositions between samples were statistically tested with the Mantel test using the mantel.rtest function in the ade4 library of R software. To identify co-association of red-snow samples obtained from the Arctic and Antarctic, we visualized the sample–OTU co-occurrence network in Cytoscape (version 3.1.0) with an edge-weighted, spring-embedded layout of 20,000 randomly picked unique sequences.

**Phylogenetic analysis**. Representative sequences of the 18S rRNA gene and ITS2 sequences were aligned using MAFFT6[32]. These alignments were carefully inspected by eye, and all ambiguous sites and sequences were manually deleted. Maximum likelihood trees were reconstructed using RAxML v.8.2.10[33] with the GTR+I+Γ model for each of the 18S rDNA and ITS2 datasets. To evaluate the confidence of the internal nodes, the bootstrap method was applied with 1000 replications using the rapid bootstrap algorithm[42]. In addition, the Bayesian phylogenetic analysis for the ITS2 dataset was carried out using MrBayes 3.2.6[43] with the GTR+I+Γ model. Two runs of four chains of Markov chain Monte Carlo iterations were performed for 1,000,000 generations, and the first 25% of trees were discarded as burn-in. The average standard deviation of split frequencies between the two runs of Markov chain Monte Carlo iterations was below 0.01, indicating convergence.

**Data availability**. The raw Illumina sequence datasets have been submitted to the DDBJ Sequence Read Archive under accession number DRA006819. The

nucleotide sequences have been deposited in DDBJ/EMBL/GenBank under the accession numbers LC371403 to LC371443, and LC381735 to LC381756. The in-house red-snow ITS2 sequence database and the nucleotide alignments of 98% sequence clustering of unique ITS2 sequences are available from http://redsnow18.paleogenome.jp/. Other relevant data supporting the findings of the study are available in this article and its Supplementary Information files, or from the corresponding author upon request.

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

## Acknowledgements

We thank the Spanish Polar Program for the support provided by the Juan Carlos I Station on Livingston Island. Project CTM2014-56473-R from the Spanish Polar Program contributed to the funding of the field work on Livingston Island. This study was supported by Grants-in-Aid for Scientific Research (no. 18H04139, 17H01854, 16H01772, 26247078, and 26241020) from the Japan Society for the Promotion of Science (JSPS). We acknowledge the 48th Japanese Antarctic Research Expedition (JARE). We thank Dr. Jun Uetake, Dr. Yukiko Tanabe and Dr. Takumi Murakami for collecting snow samples as part of their field research, and to Drs. Arwyn Edwards and Tristram Irvine-Fynn for Svalbard sampling supported by Great Britain Sasakawa Foundation funds. We would also like to thank Dr. Sota Tanaka for assistance in preparing the micromanipulator system. Computations were partially performed on the NIG (National Institute of Genetics) supercomputer at the ROIS (Research Organization of Information and Systems) National Institute of Genetics.

## Author contributions

T.S., R.M., H.M., and T.Y. designed the project. T.S. conceived the hypothesis. N.T., T.S., S.S., and F.N. collected snow samples and managed the glacier expedition. A.A. and T.S. supplied the sequences. H.M., R.M., and T.S. analysed the sequencing data. T.S., N.T.,

R.M., H.M., F.N., and T.Y. wrote the manuscript. All authors gave final approval for publication.

## Additional information

**Competing interests:** The authors declare no competing interests.

