## [Peer Review File · Nature Communications]

Reviewers' comments:

Reviewer #1 (Remarks to the Author):

This interesting study about biodiversity of psychrophilic microalgae causing red snow blooms presents, for the first time, a detailed molecular evaluation of OTU/phyloptype distribution using state of the art high-throughput protocols. Not only taking the traditional 18S marker, but considering also the more suitable but ITS2 marker and its complex secondary structure for species delimitation, makes this work to be a milestone in understanding fundamental ecological/biodiversity processes in an extremophilic habitat like polar melting snowfields. It can answer the question that a few cosmopolitan species of green algae dominate the red snow habitats, but that still much endemism has been detected. Certainly, the "everything is everywhere" theory is not valid for snow microalgae.

In a bipolar approach, the authors collected and sequenced red snow samples from glaciers in the Arctic and Antarctica. The concept of the study is fine, so is also length of the manuscript and the figures. The large supplement strongly supports the results.

The authors will have to address several issues and details for clarity and improvement of the discussion. Most of this I included in the word-file of the draft as comments or text correction suggestions.

Additionally, I have the following questions:

Can you include LM pics of the individual cells used for single cell sequencing? This would improve the value of the study.

Mid-latitude mountain ranges which also have many red snow algae (Europe, Asia, America, ...) and seasonal (temporary) snow fields based on soil or rock were not sampled. Thus, this is not a global study, but still a very nice bipolar one. 'Bipolar cosmopolitan' OTUs should also show up in temperate regions, I would expect. Please mention this somewhere.

The classical term '*Chlamydomonas nivalis*' is never mentioned. I am aware that this single-species diagnosis is not valid and taxonomic name-changes will follow, but you should at least mention it once together with a 'classical' reference.

18S OTUs from Svalbard and European Alps have been assigned to '*C. nivalis*'. In which of your groups do they fall?

Since always glaciers were sampled, did you not find any OTUs of true glacier algae like *Mesotaenium* or *Ancylonema*? Or were these phylotypes not used because of very low abundances?

Raphidonema-clade: Stibal et al. 2007 (<https://doi.org/10.1007/s00300-004-0709-y>) argue that these are soil algae blown onto snow surfaces, thus only secondary inhabitants of red snow. Could you mention that?

The Alaska-samples contain a larger amount of snow algae from the '*Chloromonadinia*'-group, which cause a rather orange or pink than red snow, and they do hardly occur at glaciers. *Chloromonas* snow algae are dominant in ephemeral snow fields which are eg. rock-based. Thus I am very surprised about this Alaska result for red snow on glaciers ...

Reviewer #2 (Remarks to the Author):

Segawa et al report a detailed study of the distribution of snow algae in the Arctic and Antarctic. The authors observed a handful of phylotypes account for the majority of sequences in the sample set suggesting these phylotypes are responsible for snow algae blooms (red snow algae blooms in particular). The authors employed internal transcribed spacer 2 sequencing to analyze phylotypes across polar samples and observed biogeography of snow algae populations in the Arctic and Antarctica.

Main comments:

Single-cell PCR is only mentioned 3 times in the main text. Its not clear why this approach was included and what the data add. Were these the same samples from Table S1? How or why was this data needed or necessary? How did it add to the study? This approach and the results need to be better integrated into the main text.

I think its worth mentioning in the main text when the samples were collected. There have been relatively few studies on snow algae composition throughout the melt season. Probably unlikely to change the results but worth considering that there may be selective differences in when certain species "bloom".

The results rely heavily on relative abundance of sequences in the data set. Were steps taken to ensure the primers were not biased to the groups recovered? Or were other steps taken to corroborate the DNA results - microscopy for instance? Perhaps 18S rRNA sequencing? Was this the idea for Single-cell PCR? How many single cells were sequenced? And the clones were used to design primers? A few details added to the methods could clear up this confusion.

Line 31: There are snow algae on glaciers and snowfields throughout the world. While the at a presented here includes the large ice sheets, it does not include any glaciers or snowfields and thus is not a "global dataset". Same comment for Line 34.

Line 38: Snow algae bloom throughout the melt season. More accurate to present them as occurring during the melt season. (And most of the samples were collected in the summer according to table S1)

Line 71: There is no evidence presented that exhaustive attempts have been made to culture. Its misleading to state they are uncultivable.

Line 83: What large number? Is not clear what "The large numbers or sequences" refers to.

Lines 83-85: How do these relate to the subgroups already defined?

Line 227: The data must be made available with a valid accession number. In addition, the ITS sequence databases should be made available (and the in-house Kolliela 18S rRNA database).

Figure 2 is a bit confusing. It would help to add to the label that these are unique sequences. And then its confusing that these are numbers but also %.

Reviewer #3 (Remarks to the Author):

This is an interesting and timely study on determining the geographical distribution of snow algal in polar regions using the ITS2 gene sequencing to obtain higher level resolution that previously obtained through culturing and 18S profiling. As indicated in the abstract, the results indicate that most snow algal phylotypes are endemic to either the Arctic or Antarctic with much fewer cosmopolitan phylotypes detected. This appears to be the major novel finding of the manuscript.

The manuscript could be improved as follows:

1. The abstract should more clearly indicate that the most significant results were obtained using ITS2 sequencing. Line 30 – are snow algae only found in thawing snow? The last sentence of the abstract is confusing and seems to contradict the major findings of the study ie that most snow algal phylotypes are endemic and not bipolar.
2. The authors need to clearly indicate the significance of this paper in a broader context beyond reporting just on the biogeographical distribution of snow algal in the polar regions. Or what is highly novel about this paper that is significant in a more global context than just to the snow algal scientific community?
3. The objective and role of using single cell PCR in this paper needs to be more clearly explained in meeting the overall objectives of the paper, the results obtained, and in the overall interpretation of the paper. In the current main manuscript, a photo is shown in Fig. 1 but no other information is really given in the actual ITS2 sequencing obtained in the main manuscript. In essence, there is no depth to the single cell PCR component to this paper in the main manuscript, although it is referred to in a much stronger fashion in the supplementary material.
4. It needs to be more clearly indicated throughout the main paper that what was actually sequenced and interpreted. Were 18S-ITS2 sequences or were the 18S and ITS2 sequences done separately as this is not clearly indicated in the main manuscript? I read over the methods section a couple of times and still could not figure out what the Authors actually did. It is implied that both 18S and ITS2 regions were both sequenced but Figures 1 and 2 only indicate ITS2 results? It was only when I went through the Supplementary Materials Section, did I finally begin to understand what the actual sequencing strategy was.

Reviewers' comments:

Reviewer #1 (Remarks to the Author):

This interesting study about biodiversity of psychrophilic microalgae causing red snow blooms presents, for the first time, a detailed molecular evaluation of OTU/phyloptype distribution using state of the art high-throughput protocols. Not only taking the traditional 18S marker, but considering also the more suitable but ITS2 marker and it's complex secondary structure for species delimitation, makes this work to be a milestone in understanding fundamental ecological/biodiversity processes in a extremophilic habitat like polar melting snowfields.

It can answer the question that a few cosmopolitan species of green algae dominate the red snow habitats, but that still much endemism has been detected. Certainly, the "everything is everywhere" theory is not valid for snow microalgae.

In a bipolar approach, the authors collected and sequenced red-snow samples from glaciers in the Arctic and Antarctica. The concept of the study is fine, so is also length of the manuscript and the figures. The large supplement strongly supports the results.

The authors will have to address several issues and details for clarity and improvement of the discussion. Most of this I included in the word-file of the draft as comments or text correction suggestions.

→ Thank you for the comments. We have revised our manuscript according to the reviewer's critique. We have explained the weakness and the limitations of our approach in more detail in the revised discussion text. Also, as suggested by the reviewer, we added sampling site information (red snow at glacier surface and non-glacier seasonal snow fields). In our original manuscript (Word file), we have replied to the comments you wrote there. Below, please find our detailed responses to your other comments.

Additionally, I have the following questions:

Can you include LM pics of the individual cells used for single cell sequencing? This would improve the value of the study.

→ Thank you for the valuable comment. In Supplementary Figure 22, we have added photos of individual cells used for single-cell PCR. We note that it is impossible to use a glass slide for taking microscope photos owing to the risk of contamination, and therefore the quality of the photos is not good, and we did not take photos for all algae cells.

Mid-latitude mountain ranges which also have many red snow algae (Europe, Asia, America, ...) and seasonal (temporary) snow fields based on soil or rock were not

sampled. Thus, this is not a global study, but still a very nice bipolar one. 'Bipolar cosmopolitan' OTUs should also show up in temperate regions, I would expect. Please mention this somewhere.

→ We agree with the reviewer. Mid-latitude mountain ranges also have many red-snow algae yet were not sampled; indeed this is one of the limitations of our study. According to the reviewer's comments, we now discuss this limitation and mention that future studies will expand the range of red-snow sampling to the mid-latitudes (lines 69–72, and 173–177).

The classical term 'Chlamydomonas nivalis' is never mentioned. I am aware that this single-species diagnosis is not valid and taxonomic name-changes will follow, but you should at least mention it once together with a 'classical' reference. 18S OTUs from Svalbard and European Alps have been assigned to 'C. nivalis'. In which of your groups do they fall?

→ We agree with the reviewer. Based on the reviewer's comments, we now mention *Chlamydomonas nivalis* with an additional reference as follows: "Several taxa of red-snow algae have been recognized in snow fields worldwide, and most have been identified based on microscopic features of the cells. Spherical red-snow cells have often been identified as *Chlamydomonas* cf. *nivalis* (Kol, 1968) and can be regarded as a cosmopolitan cryophilic species" (lines 51–54).

Also, we agree with the reviewer that 18S OTUs from Svalbard and the European Alps have been assigned to *Chlamydomonas nivalis*. We have explained "*C. nivalis*" in the revised manuscript: "The '*Chlamydomonas*'-snow group included field-collected samples from Svalbard and European Alps that had been assigned to *C. nivalis*" (lines 98–100).

Since always glaciers were sampled, did you not find any OTUs of true glacier algae like *Mesotaenium* or *Ancylonema*? Or were these phylotypes not used because of very low abundances?

→ According to previous studies, *Mesotaenium* and *Ancylonema* are glacier-ice algae, not snow algae (e.g., Takeuchi 2001 Hydrological Processes; Takeuchi 2013 Environ. Res. Lett.; and Lutz et al., 2015 Front. Microbiol.). Therefore, we reason that *Mesotaenium* and *Ancylonema* were at very low abundance in our snow samples.

In addition, the ITS sequence for *Mesotaenium* and *Ancylonema* has not been deposited in the NCBI-nt DNA sequence database. Although the BLASTN algorithm can detect low-similarity sequences, we did not find their ITS sequences in a BLASTN search with >85% similarity using the current NCBI-nt DNA sequence database.

[redacted] Therefore, *Mesotaenium* and *Ancylonema* were indeed at very low abundance in our red-snow samples.

Raphidonema-clade: Stibal et al. 2007 (<https://doi.org/10.1007/s00300-004-0709-y>) argue that these are soil algae blown onto snow surfaces, thus only secondary inhabitants of red snow. Could you mention that?

→ We appreciate this valuable comment and now cite the relevant reference. In the revised text, we note that *Raphidonema* algae are soil algae that probably were blown onto the snow surface (lines 154–158).

The Alaska-samples contain a larger amount of snow algae from the 'Chloromonadinia'-group, which cause a rather orange or pink than red snow, and they do hardly occur at glaciers. Chloromonas snow algae are dominant in ephemeral snow fields which are eg. rock-based. Thus I am very surprised about this Alaska result for red snow on glaciers ...

→ Several studies have reported that *Chloromonas* has been identified in snow samples on glaciers. For example, *Chloromonas* was identified in red snow on glaciers in Greenland, Iceland, Svalbard, and Sweden (Lineutz et al., 2016 Nature Comm.) and in supraglacial snow in Oregon and Washington, USA (Hamilton and Having, 2017 Geobiology). We now mention this fact in the revised manuscript (lines 100–102).

Reviewer #2 (Remarks to the Author):

Segawa et al report a detailed study of the distribution of snow algae in the Arctic and Antarctic. The authors observed a handful of phylotypes account for the majority of sequences in the sample set suggesting these phylotypes are responsible for snow algae blooms (red snow algae blooms in particular). The authors employed internal transcribed spacer 2 sequencing to analyze phylotypes across polar samples and observed biogeography of snow algae populations in the Arctic and Antarctica.

Main comments:

Single-cell PCR is only mentioned 3 times in the main text. Its not clear why this approach was included and what the data add. Were these the same samples from Table S1? How or why was this data needed or necessary? How did it add to the study? This approach and the results need to be better integrated into the main text.

→ We thank Reviewer #2 for their detailed and valuable comments. Because information on algae ITS2 sequences is limited at present in public DNA sequence databases, it is difficult to conduct taxonomic classification for snow algae based on general homology-based search approaches. Therefore, we constructed an in-house red-snow ITS2 sequence database in this study. For this purpose, the 18S-ITS sequences were obtained from red-snow samples via Sanger sequencing; moreover, those sequences were directly sequenced from cells morphologically identified as *C. nivalis* from the red-snow samples, and this was accomplished with single-cell PCR. This is now clearly stated in the revised manuscript (lines 79–87).

I think its worth mentioning in the main text when the samples were collected. There have been relatively few studies on snow algae composition throughout the melt season. Probably unlikely to change the results but worth considering that there may be selective differences in when certain species “bloom”.

→ We agree with the reviewer, and hence we have added information concerning the sampling season (line 181) and details of the sampling months (Supplementary Table 1). Indeed, there has been no report about changes in the community of snow algae across seasons. We assume that the community composition of red-snow algae does not change drastically during the melt season because red snow materials in this study are generally dominated by red spherical cysts which are morphologically indistinguishable from each other under light microscope.

The results rely heavily on relative abundance of sequences in the data set. Were steps taken to ensure the primers were not biased to the groups recovered? Or were other steps taken to corroborate the DNA results - microscopy for instance?

Perhaps 18S rRNA sequencing? Was this the idea for Single-cell PCR? How many single cells were sequenced? And the clones were used to design primers? A few details added to the methods could clear up this confusion.

→ We thank Reviewer #2 for these comments. We investigated taxonomic coverage of 18S rRNA gene primers and ITS primers as follows. (i) We picked 4 or 5 reference 18S rRNA gene sequences or ITS sequences from the NCBI-nt sequence database per the representative unicellular eukaryotic algae group. (ii) We carried out multiple sequence alignment with these sequences and the primer sequences. (iii) We counted mismatched nucleotides per primer-template pair. (iv) We calculated the taxonomic coverage of the primer per representative unicellular eukaryotic algae group. The results revealed that both of the primers for the 18S rRNA gene and ITS2 region used in this study exhibited good taxonomic coverage against representative snow-algae groups. Although we found an average of five nucleotide mismatches per R primer-template for the ITS2 region and found that only the *Chloroidium* group exhibited six nucleotide mismatches, almost all the mismatches are located in the 5' region (see ITS_trimmedMSA.fasta at <http://redsnow18.paleogenome.jp>), which is a usually small effect for PCR bias (Walters et al. 2011 Bioinformatics), suggesting that our primers had only a low probability of causing PCR bias among these taxa.

In addition, to estimate the bias of our ITS2 primers, we performed shotgun metagenomic sequencing with the Svalbard sample (Austre Brøggerbreen) and compared taxonomic compositions between the ITS2 amplicon sequencing data and shotgun metagenomic sequencing data. Both of taxonomic compositions were inferred using the ITS2 sequence (see AmpliconMetagenome.pdf at <http://redsnow18.paleogenome.jp>).

The taxonomic assignment results show that the amplicon sequencing data and shotgun metagenomic sequencing data for the sample yielded a similar taxonomic composition (Pearson correlation coefficient = 0.99). This result indicates that the taxonomic composition based on amplicon sequencing of ITS2 presented in this study is not biased. However, we agree with your opinion, and therefore we now show the multiple alignment files and comparison of taxonomic classification by shotgun library reads

composition and ITS2 PCR-based approach in <http://redsnow18.paleogenome.jp>. We believe that the revised manuscript clarifies this aspect, for the benefit of readers.

We apologize for the confusion concerning the single-cell PCR analysis. To obtain the sequences from algal cells morphologically identified as *C. nivalis* from the red-snow samples, we carried out single-cell PCR. In this way, we could directly obtain sequences for the 18S rRNA–ITS2 region from snow algal cells present in the red-snow samples, which were collected from Alaska and Svalbard. We used more than 20 cells in each sample for single-cell PCR. This aspect is now clearly stated in the revised manuscript (lines 79–87 and 222–241).

Line 31: There are snow algae on glaciers and snowfields throughout the world. While the at a presented here includes the large ice sheets, it does not include any glaciers or snowfields and thus is not a “global dataset”. Same comment for Line 34.

→ We agree with the reviewer that this statement was inappropriate. We have revised the text accordingly (lines 34).

Line 38: Snow algae bloom throughout the melt season. More accurate to present them as occurring during the melt season. (And most of the samples were collected in the summer according to table S1)

→ We agree with the reviewer and have revised the text accordingly (line 42).

Line 71: There is no evidence presented that exhaustive attempts have been made to culture. Its misleading to state they are uncultivable.

→ We agree with the reviewer and have revised the text accordingly to “have yet to be successfully cultured” (line 78–79).

Line 83: What large number? Is not clear what “The large numbers or sequences” refers to.

→ We agree with the reviewer and have revised the text accordingly to “Among those 22 subgroups” (lines 95–96).

Lines 83-85: How do these relate to the subgroups already defined?

→ Sorry for your confusion. These groups were classified into 22 subgroups. We have revised the manuscript to clarify this point (lines 95–98).

“Among those 22 subgroups, sequences belonged to the ‘*Chlamydomonas*’-snow group A (6%), ‘*Chlamydomonas*’-snow group B (71%), *Raphidonema* group (16%), and *Chloromonadinia*-snow group G (6%).”

Line 227: The data must be made available with a valid accession number. In addition, the ITS sequence databases should be made available (and the in-house Koliela 18S rRNA database).

→ According to the reviewer's comment, we added the accession numbers (DRA00681) and the web site address of the in-house red-snow ITS2 sequence database (<http://redsnow18.paleogenome.jp>) (lines 374–379).

Figure 2 is a bit confusing. It would help to add to the label that these are unique sequences. And then its confusing that these are numbers but also %.

→ We agree with the reviewer and have revised the figure accordingly, and we have written “% in the library of the unique sequences” in the y-axis label.

Reviewer #3 (Remarks to the Author):

This is an interesting and timely study on determining the geographical distribution of snow algal in polar regions using the ITS2 gene sequencing to obtain higher level resolution that previously obtained through culturing and 18S profiling. As indicated in the abstract, the results indicate that most snow algal phylotypes are endemic to either the Arctic or Antarctic with much fewer cosmopolitan phylotypes detected. This appears to be the major novel finding of the manuscript.

The manuscript could be improved as follows:

1. The abstract should more clearly indicate that the most significant results were obtained using ITS2 sequencing. Line 30 – are snow algae only found in thawing snow? The last sentence of the abstract is confusing and seems to contradict the major findings of the study ie that most snow algal phylotypes are endemic and not bipolar.

→ According to the reviewer's comment, we rewrote the Abstract to clearly indicate that the most significant results were obtained using ITS2 sequencing. Moreover, because this speculative part is not the main topic of our study, we decided to remove the last sentence of "globally dispersed via the atmosphere across both polar regions" that was present in the original manuscript (line 40).

2. The authors need to clearly indicate the significance of this paper in a broader context beyond reporting just on the biogeographical distribution of snow algal in the polar regions. Or what is highly novel about this paper that is significant in a more global context than just to the snow algal scientific community?

→ In accordance with the reviewer's comment, we now describe what is most pertinent about our results and how they are significant in a more global context. This description is written in the last paragraph of the main-manuscript text (lines 159–177).

3. The objective and role of using single cell PCR in this paper needs to be more clearly explained in meeting the overall objectives of the paper, the results obtained, and in the overall interpretation of the paper. In the current main manuscript, a photo is shown in Fig. 1 but no other information is really given in the actual ITS2 sequencing obtained in the main manuscript. In essence, there is no depth to the single cell PCR component to this paper in the main manuscript, although it is referred to in a much stronger fashion in the supplementary material.

→ We thank Reviewer #3 for valuable comments. We agree with the reviewer. We carried out single-cell PCR to obtain sequences for algal cells that had been morphologically identified as *C. nivalis* from the red-snow samples. Single-cell PCR was used to directly obtain sequences for the 18S rRNA–ITS2 region from snow algal cells in the red-snow samples collected from Alaska and Svalbard. In the revised manuscript, we clearly explain this aspect (lines 79–87).

4. It needs to be more clearly indicated throughout the main paper that what was actually sequenced and interpreted. Were 18S-ITS2 sequences or were the 18S and ITS2 sequences done separately as this is not clearly indicated in the main manuscript? I read over the methods section a couple of times and still could not figure out what the Authors actually did. It is implied that both 18S and ITS2 regions were both sequenced but Figures 1 and 2 only indicate ITS2 results? It was only when I went through the Supplementary Materials Section, did I finally begin to understand what the actual sequencing strategy was.

→ We thank Reviewer #3 for their detailed and valuable comments. Owing to the fact that, at present, algal ITS2 sequence data is limited in public DNA sequence databases, it is difficult to conduct taxonomic classification for snow algae based on general homology-based search approaches. Therefore, we constructed an in-house red-snow ITS2 sequence database in this study. For this purpose, the 18S-ITS sequences were obtained from red-snow samples via Sanger sequencing; moreover, those sequences were directly sequenced from cells morphologically identified as *C. nivalis* from the red-snow samples, and this was accomplished with single-cell PCR. This is now clearly stated in the revised manuscript (lines 79–87, 199–204 and 291–308).